# Echocardiography and MALDI-TOF Identification of Myosin-Binding Protein C3 A74T Gene Mutations Involved Healthy and Mutated Bengal Cats

**DOI:** 10.3390/ani12141782

**Published:** 2022-07-12

**Authors:** Kanokwan Demeekul, Pratch Sukumolanan, Chattida Panprom, Siriwan Thaisakun, Sittiruk Roytrakul, Soontaree Petchdee

**Affiliations:** 1Department of Cardio-Thoracic Technology, Faculty of Allied Health Sciences, Naresuan University, Phitsanulok 65000, Thailand; khwan_cp@hotmail.com; 2Veterinary Clinical Studies Program, Graduate School, Kasetsart University, Nakorn Pathom 73140, Thailand; pratch.vet@gmail.com; 3Kasetsart University Veterinary Teaching Hospital, Faculty of Veterinary Medicine, Kasetsart University, Nakhon Pathom 73140, Thailand; moo_breeze@hotmail.com; 4Functional Proteomics Technology Laboratory, Functional Ingredients and Food Innovation Research Group, National Center for Genetic Engineering and Biotechnology (BIOTEC), National Science and Technology Development Agency, Pathumthani 12120, Thailand; kung_siiriwan@hotmail.com (S.T.); sittiruk2000@gmail.com (S.R.); 5Department of Large Animal and Wildlife Clinical Sciences, Faculty of Veterinary Medicine, Kasetsart University, Nakorn Pathom 73140, Thailand

**Keywords:** Bengal cats, myosin-binding protein C3 A74T gene mutation, hypertrophic cardiomyopathy, echocardiography, peptidomics analysis

## Abstract

**Simple Summary:**

The most common cardiomyopathy in feline is hypertrophic cardiomyopathy (HCM) that is recognized as an inheritance of autosomal dominant transmission in some cats. Feline HCM has been associated with the mutation of the myosin-binding protein C3 (*MYBPC3*) gene in several locations with a specific breed. Recently, the proteomic analyzed by mass spectrometry is a promising novel approach to characterize the specific pattern of peptide and protein for relatively clinical biomarkers. However, there has been only limited knowledge of differential peptides and proteins related to the *MYBPC3* gene mutation. Therefore, this study aimed to investigate the *MYBPC3* gene mutation in Bengal cats with HCM. In combined with mass spectrometry, we purposed to identify the potential peptide candidates and expected proteins. The obtained findings of specific peptides and proteins from mass spectrometry may contribute to the development of novel diagnostic tools for feline HCM related to the *MYBPC3* gene mutation. Further, this observation appears to be useful information such as a database of peptidomics library in veterinary medicine for the future diagnosis of feline cardiomyopathy associated with gene mutation.

**Abstract:**

This study aimed to identify the potential peptide candidates and expected proteins associated with *MYBPC3-A74T* gene mutations in Bengal cats and determine if peptidome profiles differ between healthy controls and cats with *MYBPC3-A74T* gene mutations. All animals were evaluated using echocardiography. DNA was isolated and followed by the screening test of *MYBPC3* gene mutation. The MALDI-TOF mass spectrometry was conducted for analyzing the targeted peptide and protein patterns. The expected protein candidates were searched for within the NCBI database. Our results demonstrated that the *MYBPC3-A74T* gene mutation was dominant in Bengal cats but not in domestic shorthair cats. Correlations between baseline characteristics and echocardiographic parameters were discovered in Bengal cats. Mass spectrometry profiles of the candidate proteins were suspected to accompany the cat with the *MYBPC3-A74T* gene mutation, involving integral protein–membrane, organization of nucleus, DNA replication, and ATP-binding protein. Therefore, *MYBPC3-A74T* gene mutations occur frequently in Bengal cat populations. The high incidence of homozygotes for the mutation supports the causal nature of the *MYBPC3-A74T* mutation. In addition, peptidomics analysis was established for the first time under this condition to promise a complementary technique for the future clinical diagnosis of the *MYBPC3-A74T* mutation associated with physiological variables and cardiac morphology in cats.

## 1. Introduction

The Bengal cat breed has become popular over the past three decades. Bengal cats have attracted attention as a relative-range breed related to familial hypertrophic cardiomyopathy (HCM) [1]. HCM is the most common feline cardiac disease and is characterized by concentric hypertrophy of the left ventricular wall without signs of an underlying cause, such as pressure overload [2,3]. Feline HCM is well known as an inherited disease with the autosomal dominant transmission in some breeds, such as the Maine coon, Ragdoll, British shorthair, and Persian [4,5,6,7,8]. The recognition of two sarcomeric genes, myosin-binding protein C (*MYBPC3*) and beta-myosin heavy chain (*MYH7*), are the most essential variants in felines associated with the HCM phenotype [6,9,10,11]. Recently, the single nucleotide polymorphism (SNP) at A31P and A74T in the *MYBPC3* gene has been shown responsible for HCM in some breeds, especially Maine coon cats [1,12]. However, there are still few reports focusing on the specific gene mutation associated with HCM in Bengal cats.

In recent years, several tests have been approached as applicable investigations for cardiomyopathy in animals. Echocardiography is recommended as the gold standard method to characterize the alteration of both structure and function in the heart. In the veterinary field, this technique is preferred as a screening test for the diagnosis of HCM [2,13,14]. According to HCM related to inheritable disease, the genetic investigation is still required. Therefore, DNA sequencing remains the principal test for determining the exact nucleic acid sequence [4,15]. Certainly, many rapid screening tests have been mentioned, such as the loop-mediated isothermal amplification assay (LAMP) and the lateral flow dipstick (LFD) test [15,16,17,18]. As a result, applying the LAMP technique combined with LFD on felines with HCM will be an alternative screening test.

Advancement proteomic approaches currently become a novel instrument used in clinical diagnosis not only in humans but also in veterinary medicine due to reliable and rapid interpretation. In our earlier publication, the identification of peptides and proteins related to familial HCM was reported in Maine coon cats via matrix-assisted laser desorption ionization-time of flight (MALDI-TOF) mass spectrometry [19]. Analysis of peptides relied on unique peptide mass fingerprints (PMFs), and clusters of individual groups in HCM cats were identified using a three-dimensional principal component analysis (3D-PCA) scatterplot. Thereby, the potential biomarkers of Maine coon cats with HCM have been particularly revealed. However, little is known about peptide indicators for the diagnosis and prognosis of feline HCM according to breed, typically in Bengal cats. Therefore, this study first aimed to evaluate the phenotype and genotype of HCM in Bengal cats. In addition, this study aimed to offer the requisite information on clinical appearance, genetic variants, and candidate peptides related to feline HCM. Thus, conducting a study on Bengal cats is beneficial for novel knowledge of the diagnosis, prognosis, treatment strategies, and prevention of HCM associated with gene mutations.

## 2. Materials and Methods

### 2.1. Animals

A prospective case-control study was designed in the present study. Twenty-three cats, 9 domestic shorthair cats as control, and 14 Bengal cats were recruited from Animal Teaching Hospital Kamphaeng Saen, Faculty of Veterinary Medicine, Kasetsart University, and a private animal hospital with full owner informed consent. Before enrollment, a clinical record of each cat was accomplished by history talking and a complete physical examination. Blood sampling was collected via venipuncture, followed by centrifugation. The serum sample was separated, aliquoted into a collection tube, and stored at −20 °C before undertaking further peptidomics analysis. All cats underwent an evaluation of the cardiovascular system by using thoracic auscultation for checking their heart sound. Indirect blood pressure measurement was performed by Doppler technique using an Ultrasonic Doppler Flow Detector (Parks Medical, Beaverton, OR, USA). According to the recommendation by the ACVIM consensus statement, the blood pressure measurement was accessed by a veterinarian on a front limb of cats in lateral recumbency [14,20]. Blood pressure measurement was triplicated and represented as an average value.

### 2.2. Echocardiography

The echocardiographic examination was performed to evaluate the heart structure and heart function by one well-trained veterinary clinician with a cardiac ultrasound machine (GE, Boston, MA, USA) with continuous electrocardiographic recording. Transthoracic two-dimensional (2D), motion mode (M-mode), and Doppler mode of echocardiography were routinely assessed both in the right and left lateral recumbency of animals. Measurements were made from recorded images, as represented in Figure 1. Regarding cardiomyopathy guidelines, more than or equal to 6 mm of left ventricular wall thickness is defined as left ventricular hypertrophy [14,21]. The cats affected by valve regurgitation were excluded from the study.

### 2.3. DNA Extraction and Polymerase Chain Reaction (PCR)

DNA extraction and polymerase chain reaction (PCR) method were performed for myosin-binding protein C3 gene (*MYBPC3*) as described previously in the Appendix A [22]. Briefly, genomic DNA extraction was obtained from the frozen blood samples according to the manufacturer’s guidance (Qiagen, Hilden, Germany). Genomic DNA amplification was accomplished by specific primers, including forward primer set 5′AGCCTTCAGCAAGAAGCCA3′ and reverse primer set 5′CAAACTTGACCTTGGAGGAGC3′ [23]. The PCR reaction was conducted as previously described [18].

### 2.4. DNA Sequencing

The amplified PCR product was subsequently purified by PCR Purification Kit (Favorgen, Ping-Tung, Taiwan) for DNA sequencing. The sequencing reaction was conducted with the Sanger sequencing method, using the same specific primers as mentioned above. The DNA sequencing for *MYBPC3-A74T* and *A31P* polymorphism was analyzed with the Bioedit program.

### 2.5. MYBPC3-LAMP-LFD Detection

According to our previous study in the development of LAMP with LFD detection in *MYBPC3-A31P* gene mutation, this study applied this screening test for detecting the *MYBPC3-A31P* mutation, focusing on Bengal cats. The sets of designed *MYBPC3-A31P* LAMP primers followed the previous study protocol [24]. In this study, Maine coon cats with wild-type and mutant of *MYBPC3-A31P* were used for negative and positive control, respectively. The interpretation of the LFD test relied on the manufacturer’s instructions.

### 2.6. Serum Peptide Analysis Using MALDI-TOF MS

Serum samples were prepared with some modifications as described previously [18]. In short, the total protein concentration in the individual serum sample was evaluated by the modified Lowry protein assay [25]. The mass spectra of a particular peptide were determined by using flexAnalysis version 3.3 software (Bruker Daltonics, Bremen, Germany). While the peptide mass fingerprint (PMF), pseudo-gel view, and three-dimensional principal component analysis (3D-PCA) were investigated with ClinPro Tools version 3.0 software (Bruker Daltonics, Bremen, Germany) [26,27,28,29].

### 2.7. Identification of Peptide by LC-MS/MS

To further identify the specific peptide of merged peptide mass spectra in all sample groups, peptide samples were employed to reverse-phase high-performance liquid chromatography (HPLC). The gradient of eluted peptides was analyzed using an Ultimate 3000 LC System (Thermo Scientific Dionex, Sunnyvale, CA, USA) combined with an HCTUltra PTM Discovery System (Bruker Daltonics, Bremen, Germany). Next, peptides were scattered on a PepSwift monolithic column (100 µm internal diameter ×50 mm in length) (Thermo Fisher Scientific, Waltham, MA, USA). A linear gradient of 10–70% ACN in 0.1% formic acid (FA) was prepared. The nanocolumn system was coupled with an electrospray ionization (ESI) ion-trap mass spectrometer (Bruker Daltonics, Bremen, Germany) [29]. DeCyder MS Differential Analysis software (Amersham Biosciences, Amersham, UK) was used for peptide identification [30]. After peptide identification, the expected proteins were then recognized by searching within the NCBI database using the MASCOT version 2.2 software (Matrix Science, London, UK) [31].

### 2.8. Statistical Analysis

Data were presented as mean ± standard error of the mean (SEM). The normal distribution of the data set was conducted by the Shapiro–Wilk test. Comparison among groups was assessed for significance using one-way analysis of variance (ANOVA) followed by the Tukey post hoc test. The correlation matrix test was performed to evaluate probability metrics. GraphPad Prism 9 software was required for all statistical analyses. A *p*-value less than 0.05 was a statistically significant difference.

## 3. Results

### 3.1. The Baseline Characteristics of the Study Animals

In the present study, the enrolled animals included 23 cats: 9 domestic shorthairs (DSH) cats and 14 Bengal cats. The Bengal cats were separated into three different groups: homozygous wild-type (WT, *n* = 1), heterozygous mutation (HET, *n* = 4), and homozygous mutation (HOM, *n* = 9). As reported in Table 1, one male Bengal–WT cat represented the maximum mean of age and blood pressure, while Bengal–HET cats demonstrated the lowest blood pressure. Moreover, the highest body weight was observed in Bengal–HOM cats when compared to other groups. However, no statistically significant difference in all clinical data was found in all groups.

### 3.2. Echocardiographic Parameters of the Study Animals

The available echocardiographic parameters of all animals are given in Table 2. We found that the variables, consisting of LA, AO, LA/AO ratio, IVSd, LVIDs, and percentage of FS, were not significantly different among groups. The parameters of LVIDd, PV Vmax, AV Vmax, and IVRT were remarkably decreased in the DSH group in comparison with the Bengal–HET group, while the Bengal–HOM group considerably increased PV Vmax and IVRT when compared to those in the DSH group. The results suggested that the measurements of LVPWd, LVIDd, IVSs, and LVPWs that were importantly different in mutated Bengal cats, only one Bengal-WT cat was hypertrophic cardiomyopathic, represented by more than 6 mm of IVSd and LVPWd.

### 3.3. The Correlation Matrix of Genotype–Phenotype and Echocardiographic Parameters in Animals

In this study, we created probability matrices that demonstrated the correlations between genotype with clinical baseline and echocardiographic parameters related to HCM (Figure 2A–H). The phenotype by echocardiography and MYBPC3-A74T genotypes provided a highlight point with an increasing probability of LVPWd and LA diameter in the heterozygous A74T mutation rather than homozygous A74T mutation (0.99 vs. 0.90 and 0.53 vs. 0.37, respectively) (Figure 2A,B). In addition, the Bengal cats with an age of more than 12 months appeared to have a higher possibility to increase the echocardiographic parameters (Figure 2C,D). Conversely, the probability of percentage of FS and IVRT in Bengal cats with an age more than 12 months was lower than in Bengal cats with an age less than 12 months (0.31 vs. 0.41 and 0.27 vs. 0.58, respectively), while Bengal cats with a higher probability of LVPWd (0.97) and IVRT (0.25) represented a bodyweight of more than 3.5 kg (Figure 2E,F). Moreover, greater probabilities in the percentage of FS (0.50) and IVRT (0.72) were obtained from male Bengal cats (Figure 2G,H). According to all integrated findings, these data assumed that male Bengal cats with the heterozygous A74T mutation may associate with HCM development at the age of more than 12 months and a body weight of more than 3.5 kg.

### 3.4. LAMP-LFD Detection for MYBPC3-A31P Mutation in Study Animals

To our recently previous observations [24], LAMP combined with the LFD technique was successfully developed in the detection of MYBPC3-A31P gene polymorphism, especially in Maine coon cats. In this study, the screening test for the MYBPC3-A31P gene mutation in the recruited study animals was performed by A31P-LAMP-LFD and confirmed by agarose gel electrophoresis, as demonstrated in Figure 3. The preliminary detection illustrated that the positive band of A31P-LAMP-LFD for the MYBPC3 gene mutation was not visualized in all groups of study animals by gel electrophoresis. While mutated Maine coon cat, a positive control, noticeably displayed a control band together with the A31P-LAMP-LFD test band, this result suggested that enrolled study animals, both DSH cats and Bengal cats, did not mainly exhibit the MYBPC3-A31P gene mutation.

### 3.5. Verification of Specific SNP in MYBPC3 Gene Mutation by DNA Sequencing

The gel electrophoresis indicated the size of the MYBPC3 gene at 242 base pairs (Figure 4). Concomitantly, results of DNA sequencing revealed that DSH cats had a homozygous wild-type (G/G), not only MYBPC3-A31P polymorphism but also MYBPC3-A74T polymorphism. On the other hand, enrolled Bengal cats were specially denoted to MYBPC3-A74T polymorphism, except for only one cat with wild-type (G/G) (Figure 5). The MYBPC3-A74T mutation Bengal cats were allocated to heterozygous-A74T mutation (*n* = 4) and predominant homozygous-A74T mutation (*n* = 9). In Bengal cats with the heterozygous-A74T mutation group, two peaks of nucleotides, consisting of guanine (G) and alanine (A), were represented in the chromatogram. Whereas, another type of mutated A74T mutation in Bengal cats was expressed by alanine (A). Accumulating observation indicated that the group of Bengal cats in this study primarily related to the MYBPC3-A74T gene mutation.

### 3.6. Results of Peptide Analysis Using MALDI-TOF

The results of mass spectrometry analyzed by the MALDI-TOF method are exhibited in Figure 6 and Figure 7. The 3D-PCA scatterplot individually distinguished the 17 selected animals into different three groups, depending on the characteristic of hypertrophic cardiomyopathy. The control group was represented by a DSH cat with homozygous wild-type. In addition, both mutated Bengal cat groups had no development of hypertrophic cardiomyopathy, except for only one Bengal cat with homozygous A74T wild-type that displayed the characteristics of cardiac hypertrophy. Peptide mass fingerprint (PMF) of study animals was conducted in the range 2000–15,000 DA among selected samples. The different mass spectral peaks of peptides were demonstrated by ClinPro Tools software. The coordinated peptides in all groups were preferred to further evaluate for peptide identification.

### 3.7. Peptide Identification and Protein Target

The selected peptides were differently identified regarding the peak of peptide mass. Additionally, the expected proteins were searched for within the NCBI database. Table 3 represented the identification of peptide mass and expected proteins. A mass spectral peak of the peptide at 3199 Da associated with the integral component membrane and zinc/metal ion binding was found in DHS cats and mutated Bengal cats. In all recruited Bengal cats, the peak of peptide mass at 916 Da was related to nuclear structural proteins, although a mass spectral peak of the peptide at 804/805 Da was not observed in Bengal cats with a heterozygous A74T mutation group. However, this peptide was correlated to adaptor protein (AP) complexes and ATP-binding. Moreover, a mass spectral peak of the peptide at 808 Da was only exhibited in Bengal cats with a heterozygous A74T mutation group. This identified peptide is linked to DNA structure and genetics.

## 4. Discussion

This study focused on illustrating the effect of *MYBPC3-A74T* gene mutations in Bengal cat populations. The present finding demonstrated that the Bengal cat predominantly carried the *MYBPC3-A74T* mutation (Figure 5). The results from DNA sequencing indicated that a single amino acid changed from guanine to adenine (G/A) in Bengal cats with a heterozygous A74T mutation, whereas a homozygous A74T mutation was represented by the replacement of an amino acid from adenine (A/A). Similarly, our study showed that Bengal cats with heterozygous *MYBPC3-A74T* gene mutations usually lack echocardiographic evidence of HCM. Moreover, homozygous *MYBPC3-A74T* gene mutations initially resulted in HCM development in young- to middle-aged cats. The results from this study and previous studies suggest that echocardiographic screening in young-aged cats should not be the sole diagnostic to identify HCM [14].

In a subsequent study of proteomic methods, the peptidomic profiles exhibited in Bengal cats compared to DSH cats may contribute to the optional methods for early detection and evaluation of HCM in cats. Peptide analysis by MALDI-TOF mass spectrometry is a rapid, accurate, and reliable technique with high sensitivity, resulting in an appropriate clinical diagnostic tool. This technique required a small volume of serum samples for analyzing several types of peptides at a time. For this reason, peptide mass derived from MALDI-TOF contains a greater potential for the diagnosis of disease. Apart from this, analysis of the 3D-PCA scatterplot by MALDI-TOF certainly separated the clusters of peptide expression among the sample groups. Based on a previous study, peptidomics were used to distinguish the proteins that correlated with cardiac hypertrophy in cats with *MYBPC3* gene mutations [18]. In agreement with the present finding, the candidate peptides and expected proteins were predominantly demonstrated in cats with a sarcomeric *MYBPC3-A74T* gene mutation (Figure 6 and Figure 7). The co-occurrence of peptides in all groups was searched for within the NCBI database to identify the expected proteins. The expected proteins consisted of an integral protein–membrane, proteins related to nucleus organization, ATP-binding protein, adaptor protein (AP) complexes, and DNA replication (Table 3). These proteins have been reported to be involved in several basic events in cellular biological processes, similar to our present study.

According to *MYBPC3* gene mutations related to HCM, calcium-binding proteins have been mostly reported for many ion-binding proteins [32]. It has been revealed that patients with a deficit of mitochondrial copper-binding proteins eventually die from fatal HCM [33]. Consistently, mutations in a protein involved in copper chaperone trafficking to cytochrome c oxidase, the terminal enzyme of the respiratory chain, lead to HCM [34,35]. Combined with a previous study confirmed by a trypsin accessibility assay, the results suggest that the soluble zinc-binding protein is important in redox regulation in the intermembrane space (IMS) of mitochondria [36,37]. However, there is still a lack of data on the promising correlation of zinc-mediated HCM.

A previous study demonstrated that the value of wall thickness above 6 mm may identify cats with left ventricular hypertrophy and HCM. This cutoff of 6 mm is higher than the upper 95% prediction interval for cats weighing <6 kg [38]. 

The echocardiographic data in the current study demonstrated that left ventricular wall thickness might associate with the *MYBPC3-A74T* mutation in Bengal cats. The values of the interventricular septum and left ventricular wall were higher in cats with homozygous for the *MYBPC3-A74T* mutation than the cat with heterozygous. On the other hand, wild-type Bengal cats represented left ventricular hypertrophy. 

Interestingly, cats with the homozygous *MYBPC3-A74T* mutation had a significant difference in the isovolumetric relaxation time compared to the control group. An explanation for the prolongation of the relaxation time could be the changes in myocardial elastic element content. The finding of expected proteins in this study agrees with a previous study of humans with HCM, which found that these proteins are involved in genetic processes. Mutation of human β-myosin disturbed a wide range of myosin molecular activities, reflecting the steps of ATP-binding and hydrolysis [39]. Moreover, ATP is well known as an energy molecule that is required for several biological events, including cardiac contraction. This situation consumes ATP for the cross-bridge cycle to generate muscle tension and/or muscle contraction [40]. However, the established role of the *MYBPC3* mutation in cardiac function is still far from complete.

The present study identifies novel peptide candidates associated with the *MYBPC3-A74T* mutation in Bengal cats. However, these peptides should be studied further to provide a better diagnosis and prognosis of the disease and for molecular-targeted treatment in the future.

## 5. Limitations

A limitation of the present study is that only a small number of Bengal owner groups are dedicated to the research. Therefore, the study animals are close to the same breed. In addition, the average age of the cat populations in this study is young; the physiological variables tracking these cat populations might be useful information for the further clinical relevance of A74T polymorphism in Bengal cats. This study did not measure the heart rate of animals. Moreover, we did not perform the experiments such as Western blot for validating the protein expressions in study animals. In addition, we are unable to control the environment and nutrition in these cat populations, which is a limitation of our study.

## 6. Conclusions

The present study revealed that the Bengal cats demonstrated a high frequency of *MYBPC3* mutations at the A74T location. Additionally, for the first time, this study established a peptidomics analysis in Bengal cats associated with the *MYBPC3-A74T* mutation. The findings for specific peptides and proteins from MALDI-TOF mass spectrometry may contribute to the development of diagnostic tools for feline HCM related to gene mutation. Further, this observation appears to be useful information for peptidomics library in veterinary medicine for the future diagnosis of feline cardiomyopathy associated with gene mutation.

## Figures and Tables

**Figure 1 animals-12-01782-f001:**
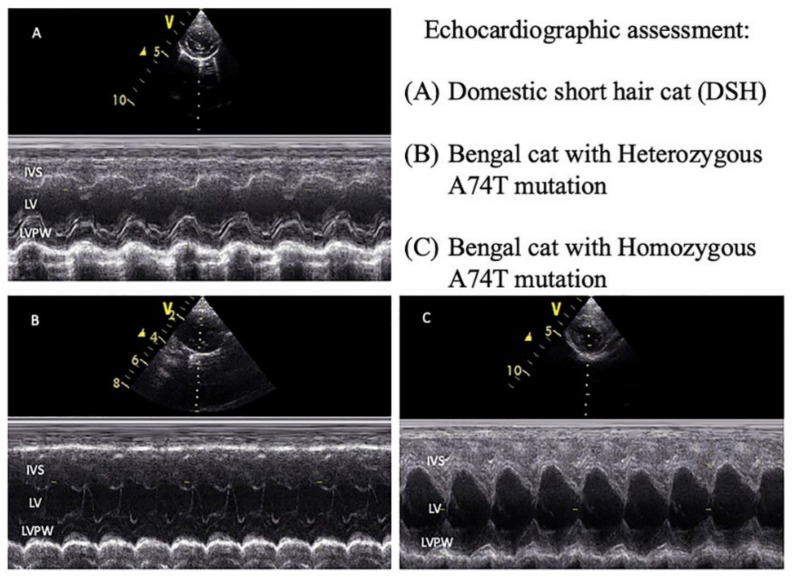
Echocardiographic assessment of left ventricular wall in the normal cat. The representative figures were visualized by the motion mode (M mode) at the papillary muscle view in the short-axis view. (**A**) DSH cats, (**B**) heterozygous A74T mutation in Bengal cats, and (**C**) homozygous A74T mutation in Bengal cats. IVS = interventricular septum; LV = left ventricle; LVPW = left ventricular proximal wall. Schemes follow the same formatting, V represents the marker position.

**Figure 2 animals-12-01782-f002:**
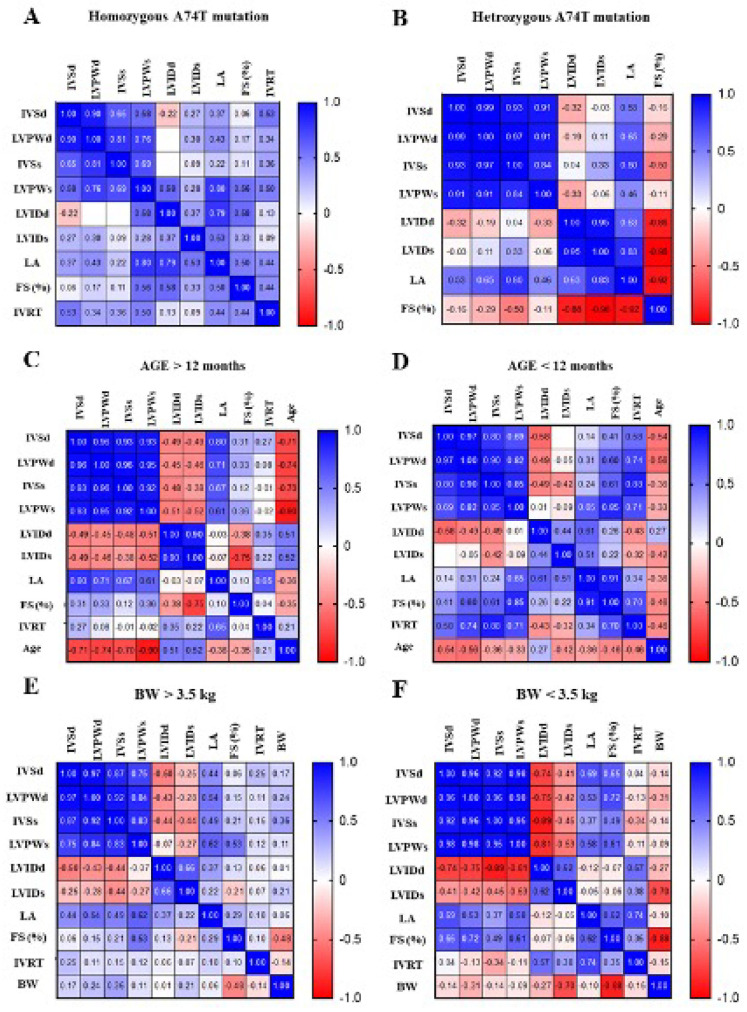
The correlation matrix of baseline characteristics and echocardiography among groups of Bengal cats with MYBPC3-A74T gene mutation. HET = heterozygous mutation; HOM = homozygous mutation; LA = left atrium; AO = aorta; IVSd = interventricular septal at end-diastole; LVPWd = left ventricular free proximal wall diameter at end-diastole; LVIDd = left ventricular internal diameter at end-diastole; IVSs = interventricular septal at end-systole; LVPWs = left ventricular free proximal wall diameter at end-systole; LVIDs = left ventricular internal diameter at end-systole; FS = fractional shortening; PV Vmax = pulmonary valve maximum blood velocity; AV Vmax = aortic valve maximum blood velocity; FS = fractional shortening; IVRT = isovolumic relaxation time.

**Figure 3 animals-12-01782-f003:**
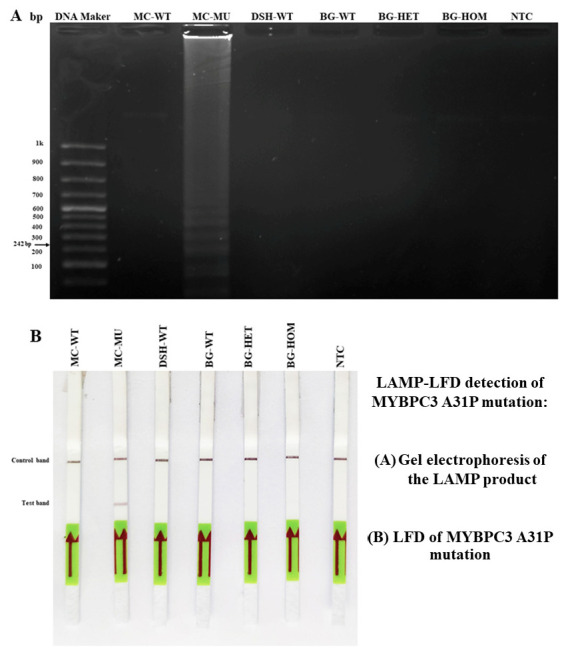
The results of LAMP-LFD detection of MYBPC3-A31P mutation in the study animals. (**A**) The determination of the LAMP product was based on the analysis of gel electrophoresis at 242 bp. (**B**) LFD of MYBPC3-A31P mutation. MC = Maine coon cat; DSH = Domestic short hair; BG = Bengal cat; WT = wild-type; MU = mutant; HET = heterozygous mutation; HOM = homozygous mutation; NTC = negative control; bp = base pair.

**Figure 4 animals-12-01782-f004:**
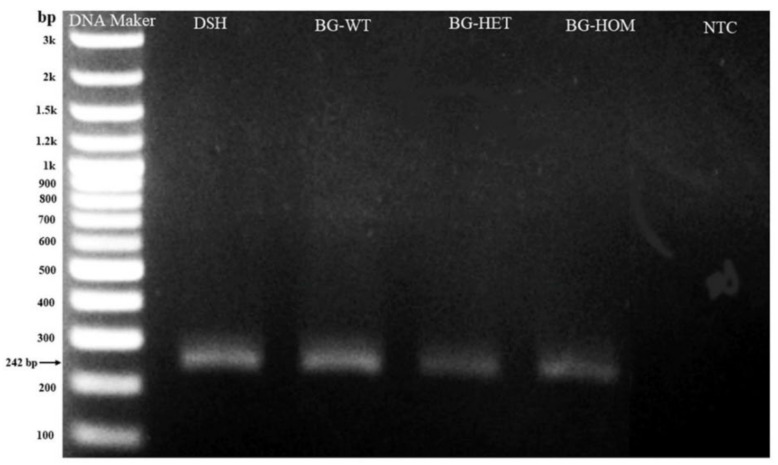
The results of gel electrophoresis. The size of the MYBPC3 gene from the PCR product was 242 bp. DSH = Domestic short hair; BG = Bengal cat; WT = wild-type; HET = heterozygous mutation; HOM = homozygous mutation; NTC = negative control; bp = base pair.

**Figure 5 animals-12-01782-f005:**
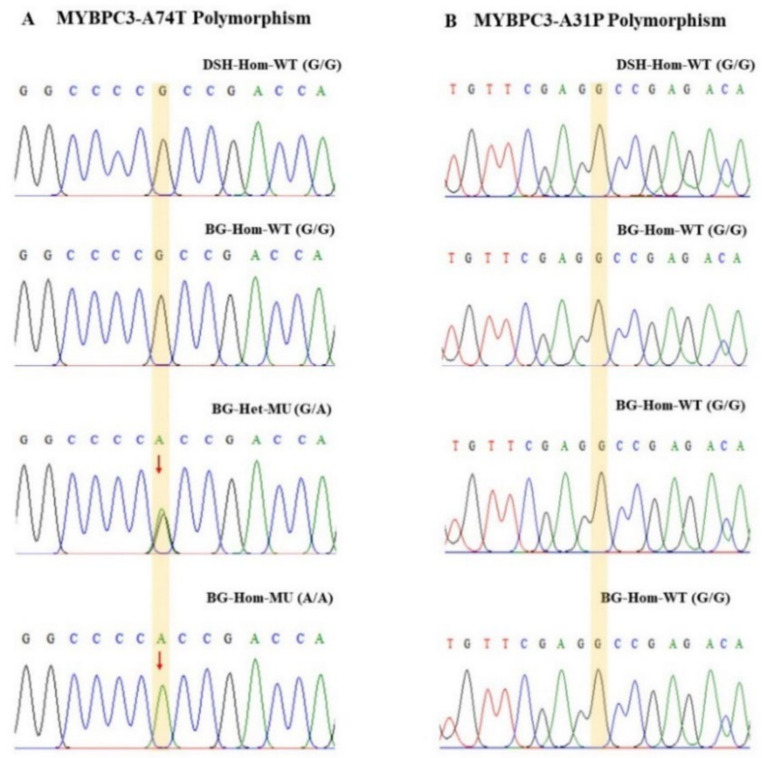
Sequence of MYBPC3 mutation in the study animals: (**A**) MYBPC3-A74T polymorphism; (**B**) MYBPC3-A31P polymorphism. DSH = Domestic short hair; BG = Bengal cat; WT = wild-type; HET = heterozygous mutation; HOM = homozygous mutation.

**Figure 6 animals-12-01782-f006:**
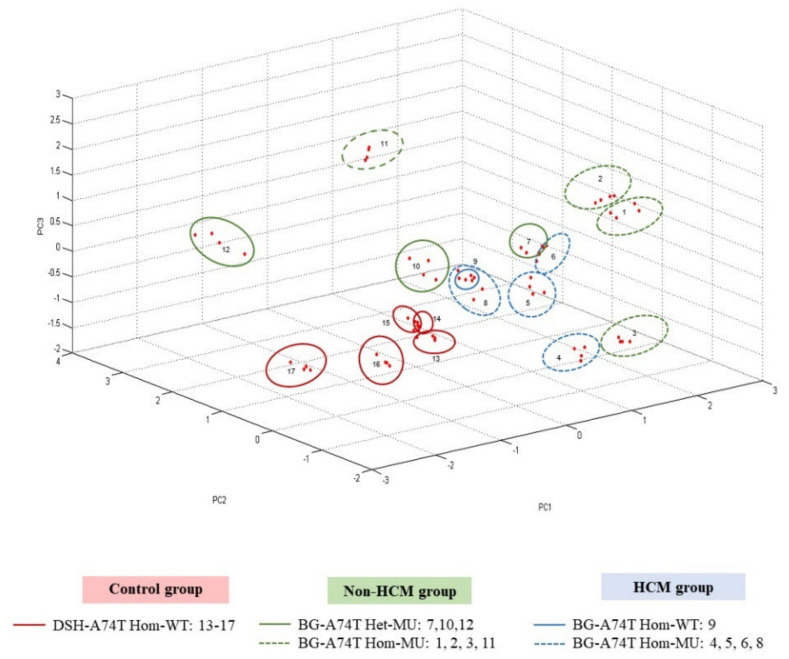
The analysis of MALDI-TOF mass spectrometry. A three-dimensional principal component analysis scatterplot (3D-PCA) of the control group, non-HCM group, and HCM group in each pattern of MYBPC3-A74T gene mutation. HCM = hypertrophic cardiomyopathy; DSH = Domestic short hair; BG = Bengal cat; WT = wild-type; MU = mutant.

**Figure 7 animals-12-01782-f007:**
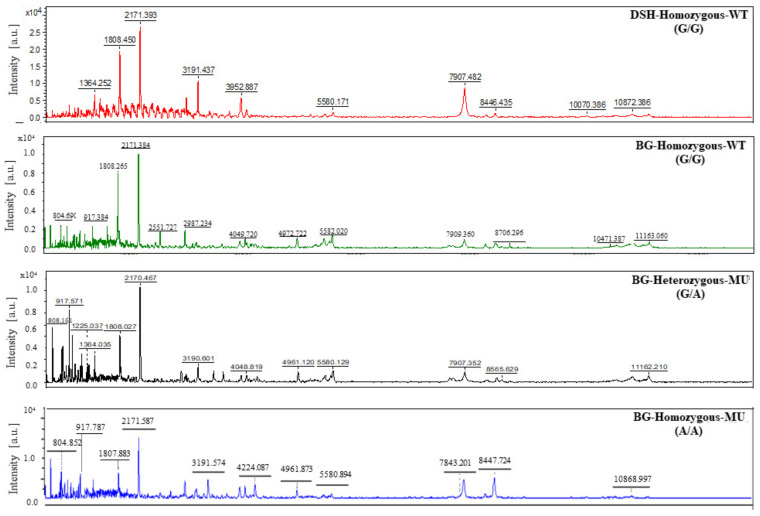
Peptide mass fingerprint (PMF) of the isolated protein from the serum of the study animals. DSH = Domestic short hair; BG = Bengal cat; WT = wild-type; MU = mutant.

**Table 1 animals-12-01782-t001:** Clinical characteristics of the study animals.

Parameters	DSH (*n* = 9)	Bengal			*p*-Value
		A74T-WT (*n* = 1)	A74T-HET (*n* = 4)	A74T-HOM (*n* = 9)	
Age (months)	20.00 ± 4.60	29.00 ± 0.00	23.63 ± 9.08	14.11 ± 2.43	0.47
Body weight (kg)	4.41 ± 0.49	4.40 ± 0.00	3.58 ± 0.35	4.89 ± 0.58	0.56
Male (number (%))	3/9 (33.33%)	0/1 (0%)	1/4 (25.00%)	3/9 (33.33%)	-
Blood pressure (mmHg)	111.10 ± 8.24	126.67 ± 0.00	95.83 ± 6.29	108.52 ± 8.20	0.60

DSH = domestic short hair; WT = wild-type; HET = heterozygous mutation; HOM = homozygous mutation.

**Table 2 animals-12-01782-t002:** Echocardiographic parameters of the study animals.

Parameters	DSH (*n* = 9)	Bengal		
		A74T-WT (*n* = 1)	A74T-HET (*n* = 4)	A74T-HOM (*n* = 9)
LA (cm)	0.96 ± 0.04	1.08 ± 0.00	0.90 ± 0.00	1.02 ± 0.04
AO (cm)	0.78 ± 0.04	0.73 ± 0.00	0.72 ± 0.02	0.70 ± 0.04
LA/AO ratio	1.30 ± 0.07	1.47 ± 0.00	1.29 ± 0.01	1.44 ± 0.06
IVSd (cm)	0.48 ± 0.03	0.61 ± 0.00	0.40 ± 0.06	0.57 ± 0.03
LVPWd (cm)	0.46 ± 0.03	0.61 ± 0.00	0.40 ± 0.05	0.57 ± 0.03 *
LVIDd (cm)	1.21 ± 0.07	1.33 ± 0.00	1.65 ± 0.02 ^##^	1.31 ± 0.06 *
IVSs (cm)	0.59 ± 0.02	0.75 ± 0.00	0.51 ± 0.06	0.70 ± 0.03 **
LVPWs (cm)	0.58 ± 0.04	0.65 ± 0.00	0.49 ± 0.08	0.69 ± 0.03 *
LVIDs (cm)	0.82 ± 0.07	0.78 ± 0.00	1.01 ± 0.04	0.79 ± 0.03
FS (%)	40.13 ± 3.88	41.30 ± 0.00	39.06 ± 1.51	41.93 ± 1.67
PV Vmax (m/s)	0.63 ± 0.02	0.76 ± 0.00	0.98 ± 0.07 ^#^	1.12 ± 0.08 ^††^
AV Vmax (m/s)	0.63 ± 0.02	07.00 ± 0.00	0.86 ± 0.06 ^##^	0.73 ± 0.03
IVRT (ms)	39.00 ± 0.24	50.00 ± 0.00	50.00 ± 0.00 ^#^	51.67 ± 3.12 ^†^

* *p* < 0.05, ** *p* < 0.01 when HET vs. HOM, ^#^ *p* < 0.05, ^##^
*p* < 0.01 when DSH vs. HET, ^†^ *p* < 0.01, ^††^
*p* < 0.0001 when DSH vs. HOM. DSH = domestic short hair; WT = wild-type; HET = heterozygous mutation; HOM = homozygous mutation; LA = left atrium; AO = aorta; IVSd = interventricular septal at end-diastole; LVPWd = left ventricular free proximal wall diameter at end-diastole; LVIDd = left ventricular internal diameter at end-diastole; IVSs = interventricular septal at end-systole; LVPWs = left ventricular free proximal wall diameter at end-systole; LVIDs = left ventricular internal diameter at end-systole; FS = fractional shortening; PV Vmax = pulmonary valve maximum blood velocity; AV Vmax = aortic valve maximum blood velocity; FS = fractional shortening; IVRT = isovolumic relaxation time.

**Table 3 animals-12-01782-t003:** Identification of mass spectral peak of peptides in the study animals.

Peptide Mass (Da)	Sample Group	Expected Protein
3199	a, c, d	Integral component membrane/Zinc ion-binding/Metal ion-binding
916	b, c, d	Guanyl-nucleotide exchange factor activity Structural constituent of nuclear pore
804/805	b, d	ATP-binding Adaptor protein (AP)-2 adapter complex binding
808	c	DNA-binding transcription factor activity Ubiquitin protein ligase binding Growth factor activity DNA clamp loader activity/Single-stranded DNA helicase activity

a—indicates DSH-Homozygous-WT (*n* = 5), b—indicates BG-Homozygous-WT (*n* = 1), c—indicates BG-Heterozygous-MU (*n* = 3), d—indicates BG-Homozygous-MU (*n* = 8). DSH = Domestic short hair; BG = Bengal cat; WT = wild-type; HET = heterozygous mutation; HOM = homozygous mutation; HCM = hypertrophic cardiomyopathy.

## Data Availability

The datasets generated and/or analyzed during the current study are available in the NCBI database repository (myosin-binding protein C3 in domestic cat; gene ID: 101094698), https://www.ncbi.nlm.nih.gov/gene/101094698.

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
