# Peer review of "Echocardiography and MALDI-TOF Identification of Myosin-Binding Protein C3 A74T Gene Mutations Involved Healthy and Mutated Bengal Cats"

_animals, 2022, doi:10.3390/ani12141782_

Round 1

Reviewer 1 Report

Reviewers comments

Echocardiography and MALDI-TOF identification of myosin binding protein C3 A74T gene mutations in Bengal cats

In general

This represents a real interesting study. The analysis methods shown here are potentially were useful for the future, especially in the population of Bengal cats. It seems that there is a population of cats with this special mutation.

The main problem from my point of view is that the analytic aspect is very good, but the correlation with the echocardiography is not realistic. It is postulated that there is a correlation to the degree of wall thickness, but only one cat is really thickened, the other ones are only in the upper normal range. So the relationship and the discussion about that must be done more specifically and details. Please rewrite these paragraphs. Beside this sometimes it is not completely clear which mutation type is discussed or analyzed. Please correct this also.

From my point of view this is a study which should be published, but there is currently work to do and so a major revision is necessary.

Abstract

Line 34: Please correct the term have performed is in this case not correct.

Introduction

This is an adequate review of the current literature introducing the problem in the cat population.

Line 68: These two references are not ideal for this statement, that the echo is the most common performed screening test in cats. The ACVIM consensus statement would be currently the best reference. Please insert this one.

Material and methods

Animals

Line 100: Please correct the term “general heart sound”.  Normally there are two heart sounds. Better in this case would be only result of the auscultation.

Line 103: the blood measurement, please insert pressure à blood pressure measurement

Echocardiography

Line 113: Please correct it is 6.0mm not only > 6mm

Figure 1:

Please correct the headline of the figure, because the first picture should be a normal cat and not an HCM cat.

Also please correct this is not a long-axis view in these pictures, it is a short a axis view.

DNA Extraction and Polymerase chain reaction

Line 141:

I am sorry you only write in here about the A31P mutation, in the title and in the abstract you write about the A74T mutation, please specify or correct

Results

The correlation matrix of genotype-phenotype and echocardiographic parameters in animals

I am sorry, but this statistical analysis and the results are only theoretical. There is only one cat with a hypertrophy and then to postulate that some cats may develop an HCM is not realistic. You did this postulation in line 221 and that is not realistic. Please specify this in another way.

Line 224

Is this really the A31P mutation?

Discussion

Line 300:

You write here that the aim of the study was to look for both mutations in Bengal cats. In the study design this is not written down. Please correct and specify this in the paragraph above.

Line 308:

This again is a postulate which is not proofed with the study results. Again there is only one cat with a hypertrophy and then the statement, that a HCM is possible in these cats is not really true. You may speculate that there are differences but this is not a sign for a future HCM. Please discuss it in more detail.

Line 355:

The following paragraph must be changed. You write that the the septum and the wall of the left ventricle were significantly thickened. That is not true. Like you write the cut off value for a hypertrophy is 6mm and then it is not possible to write this! Please specify and correct it.

What is the meaning of a larger internal diameter?

Line 360:

If you discuss the IVRT and a prolongation it would be necessary to talk about diastolic dysfunction and the degree of a diastolic dysfunction. On the other hand if I look on the values of the IVRT all are in the normal range.

Author Response

Dear Editor,

We have provided a point-by-point reply to each point raised by the reviewers as a separate file and indicated any changes or corrections within the text highlighted in yellow on the resubmitted manuscript. Please see below for the list of changes.

List of changes

Reviewer 1 Comments

In general

This represents a real interesting study. The analysis methods shown here are potentially were useful for the future, especially in the population of Bengal cats. It seems that there is a population of cats with this special mutation.

The main problem from my point of view is that the analytic aspect is very good, but the correlation with the echocardiography is not realistic. It is postulated that there is a correlation to the degree of wall thickness, but only one cat is really thickened, the other ones are only in the upper normal range. So the relationship and the discussion about that must be done more specifically and details. Please rewrite these paragraphs. Beside this sometimes it is not completely clear which mutation type is discussed or analyzed. Please correct this also.

Answer: We appreciate the helpful comments. We have edited the revised manuscript to update this point. We hope that this change has clarified this point (Line 354-358).

From my point of view this is a study which should be published, but there is currently work to do and so a major revision is necessary.

Abstract

Line 34: Please correct the term have performed in this case not correct.

Answer: We apologize for the mistake on this point. We have edited the manuscript to correct this point (Line 34-35).

Introduction

This is an adequate review of the current literature introducing the problem in the cat population.

Line 68: These two references are not ideal for this statement, that the echo is the most common performed screening test in cats. The ACVIM consensus statement would be currently the best reference. Please insert this one.

Answer: We appreciate the helpful comments. We have edited the revised manuscript to update this point (Line 99-101).We appreciate the reviewer’s understanding.

Material and methods

Animals

Line 100: Please correct the term “general heart sound”.  Normally there are two heart sounds. Better in this case would be only result of the auscultation.

Answer: We apologize for the unclear on this point. We have edited the revised manuscript to update this point (Line 69).We appreciate the reviewer’s understanding.

Line 103: the blood measurement, please insert pressure à blood pressure measurement

Answer: We apologize for the mistake at this point. We have edited the revised manuscript to update this point (Line 102-104).

Echocardiography

Line 113: Please correct it is ≥ 6.0mm not only > 6mm

Answer: We apologize for the mistake at this point. We have edited the revised manuscript to update this point (Line 112-114).

Figure 1:

Please correct the headline of the figure, because the first picture should be a normal cat and not an HCM cat.

Also please correct this is not a long-axis view in these pictures, it is a short-axis view.

Answer: We apologize for the mistake at this point. We have edited the figure’s title name in the revised manuscript to update this point. We hope that this change has clarified this point (Line 119-121).

DNA Extraction and Polymerase chain reaction

Line 141:

I am sorry you only write here about the A31P mutation, in the title and in the abstract, you write about the A74T mutation, please specify or correct

Answer: We apologize for the unclear on this point. In this study, we aimed to study MYBPC3-A74T. However, the mutation of the MYBPC3-A31P gene is the crucial SNP gene mutation. Therefore, we ruled out the A31P mutation with a screening test, A31P-LAMP-LFD, and confirmed with PCR and DNA sequencing. In this study, we have preliminarily performed A31P-LAMP-LFD as a screening test for MYBPC3-A31P mutation in Bengal cats. In addition, we have confirmed this result with the standard method, including PCR and DNA sequencing. Therefore, we did not mention this point in the title and abstract. We appreciate the reviewer’s understanding.   

Results

The correlation matrix of genotype-phenotype and echocardiographic parameters in animals

I am sorry, but this statistical analysis and the results are only theoretical. There is only one cat with a hypertrophy and then to postulate that some cats may develop an HCM is not realistic. You did this postulation in line 221 and that is not realistic. Please specify this in another way.

Answer: We apologize for the unclear on this point. We have edited the revised manuscript to update this point (Line 221-223). We appreciate the reviewer’s understanding.

Line 224

Is this really the A31P mutation?

Answer: We apologize for the unclear on this point. As we mentioned earlier, we performed the screening test for MYBPC3-A31P mutation by using A31P-LAMP-LFD in Bengal cats in this study. We hope that this change has clarified this point.   

Discussion

Line 300:

You write here that the aim of the study was to look for both mutations in Bengal cats. In the study design this is not written down. Please correct and specify this in the paragraph above.

Answer: We apologize for the unclear on this point. We have edited the revised manuscript to update this point (Line 300-302). We appreciate the reviewer’s understanding.

Line 308:

This again is a postulate which is not proofed with the study results. Again there is only one cat with a hypertrophy and then the statement, that a HCM is possible in these cats is not really true. You may speculate that there are differences but this is not a sign for a future HCM. Please discuss it in more detail.

Answer: We apologize for the unclear on this point. We have edited the revised manuscript to update this point (Line 307-308). We appreciate the reviewer’s understanding.

Line 355:

The following paragraph must be changed. You write that the septum and the wall of the left ventricle were significantly thickened. That is not true. Like you write the cut off value for a hypertrophy is 6mm and then it is not possible to write this! Please specify and correct it.

Answer: We apologize for the unclear on this point. We have edited the revised manuscript to update this point (Line 354-361). We appreciate the reviewer’s understanding.

Line 360:

If you discuss the IVRT and a prolongation it would be necessary to talk about diastolic dysfunction and the degree of a diastolic dysfunction. On the other hand if I look on the values of the IVRT all are in the normal range.

Answer: We apologize for the unclear on this point. In this study, the value of IVRT is in the normal range. However, the Bengal cat with homozygous A74T mutation represented the highest values. We have edited the revised manuscript to update this point (Line 362-364). We appreciate the reviewer’s understanding.

Best Regards

Soontaree Petchdee

Reviewer 2 Report

The title should be amended. The title should state that the study involved healthy cats.

Add the information about ethical aproval.

Add information about heart rate in table 2.

Only one cat was a homozygous wild type and data from echocardiography exam indicates fenotype HCM (IVSd=0.61, PWd=0.61). This results was enroled from statistical analysis. Why? In my opinion this results change the conlusions. The senstence "The echocardiographic data in current study demonstrate that the left ventricular wall thickness was associated with the MYBPC3-A74T mutation in Bengal cats" is not true in the light of the presented result of the cat with the wild phenotype and not included in the statistical analysis. In my opinion the authors should add this cat to statistical analysis or increase the group of wilde fenotype cats.

Author Response

Dear Editor,

We have provided a point-by-point reply to each point raised by the reviewers as a separate file and indicated any changes or corrections within the text highlighted in yellow on the resubmitted manuscript. Please see below for the list of changes.

Reviewer 2 Comments

The title should be amended. The title should state that the study involved healthy cats.

Answer: We appreciate the helpful comments. We have edited the revised manuscript to update this point. We appreciate the reviewer’s understanding.

Add the information about ethical approval.

Answer: We appreciate the helpful comments. We have stated this point in the revised manuscript ( Line 401-403). We appreciate the reviewer’s understanding.

Add information about heart rate in table 2.

Answer: We appreciate the helpful comment. In this study, we examined the heart sound by stethoscope under a trained veterinary clinician. Nevertheless, there are some limitations such as the information on heart rate. We have edited the revised manuscript to update this point (Line 382-383). We appreciate the reviewer’s understanding.

Only one cat was a homozygous wild type and data from echocardiography exam indicates fenotype HCM (IVSd=0.61, PWd=0.61). This results was enroled from statistical analysis. Why? In my opinion this results change the conlusions. The senstence "The echocardiographic data in current study demonstrate that the left ventricular wall thickness was associated with the MYBPC3-A74T mutation in Bengal cats" is not true in the light of the presented result of the cat with the wild phenotype and not included in the statistical analysis. In my opinion the authors should add this cat to statistical analysis or increase the group of wilde fenotype cats.

Answer: We appreciate the helpful comment. In this study, we attempt to identify the potential peptide candidates and expected proteins associated with MYBPC3-A74T gene mutations, focusing on Bengal cats. As we mentioned in this study, only a small number of Bengal owner groups are dedicated to the research. Therefore, the study animals are close to the same breed. Therefore, the number of animals per group might be the limitation of this study (Line 378-379). We appreciate the reviewer’s understanding.

Best Regards

Soontaree Petchdee

Round 2

Reviewer 1 Report

Reviewers comments

Echocardiography and MALDI-TOF identification of myosin binding protein C3 A74T gene mutations involved healthy and mutated Bengal cats

In general

Thanks to the authors, the manuscript was worked up in a very good way and all questions were answered sufficiently. From my point of view it is now a study, which be published and no more changes are necessary.

Again thanks to the authors for their work.

Reviewer 2 Report

Accept